# Effect of Co_3_O_4_ Nanoparticles on Improving Catalytic Behavior of Pd/Co_3_O_4_@MWCNT Composites for Cathodes in Direct Urea Fuel Cells

**DOI:** 10.3390/nano11041017

**Published:** 2021-04-16

**Authors:** Nguyen-Huu-Hung Tuyen, Hyun-Gil Kim, Young-Soo Yoon

**Affiliations:** 1Department of Materials Science and Engineering, Gachon University, Seongnam 13120, Korea; nguyenhhtuyen@gmail.com; 2ATF Technology Development Division, Korea Atomic Energy Research Institute, Daejeon 34057, Korea

**Keywords:** Pd nanoparticles, noble metals, oxygen reduction reaction, transitional metal oxide, non-Pt catalysts, cathodic urea fuel cell catalysts

## Abstract

Direct urea fuel cells (DUFCs) have recently drawn increased attention as sustainable power generation devices because of their considerable advantages. Nonetheless, the kinetics of the oxidation-reduction reaction, particularly the electrochemical oxidation and oxygen reduction reaction (ORR), in direct urea fuel cells are slow and hence considered to be inefficient. To overcome these disadvantages in DUFCs, Pd nanoparticles loaded onto Co_3_O_4_ supported by multi-walled carbon nanotubes (Pd/Co_3_O_4_@MWCNT) were employed as a promising cathode catalyst for enhancing the electrocatalytic activity and oxygen reduction reaction at the cathode in DUFCs. Co_3_O_4_@MWCNT and Pd/Co_3_O_4_@MWCNT were synthesized via a facile two-step hydrothermal process. A Pd/MWCNT catalyst was also prepared and evaluated to study the effect of Co_3_O_4_ on the performance of the Pd/Co_3_O_4_@MWCNT catalyst. A current density of 13.963 mA cm^−2^ and a maximum power density of 2.792 mW cm^−2^ at 20 °C were obtained. Pd/Co_3_O_4_@MWCNT is a prospectively effective cathode catalyst for DUFCs. The dilution of Pd with non-precious metal oxides in adequate amounts is economically conducive to highly practical catalysts with promising electrocatalytic activity in fuel cell applications.

## 1. Introduction

Recently, there has been a significant surge in attention paid to energy resources, especially renewable energy. The need to find a source of clean sustainable energy in the near future is becoming imperative. One of the best candidates is fuel cells, which do not produce any pollutants as byproducts [1,2]. Urea or urine, as a hydrogen carrier, can be used for global electricity generation, because these substances are abundant with high energy density (16.9 MJ L^−1^ in the liquid state) and are non-flammable, non-toxic, and biodegradable. In particular, urea and urine fuels are popularly used as low-cost resources, which are prominent for large-scale use in renewable energy. Moreover, they are safe for storage and transportation in the long run [3]. Over the past decade, the direct urea fuel cell (DUFC), as a sustainable power generation device, has been extensively studied because of these undeniable strengths. Nonetheless, the kinetics of the oxidation-reduction reactions, particularly electrochemical oxidation and oxygen reduction reactions, in direct urea fuel cells are sluggish; hence, improving the efficiency and performance of these reactions by utilizing highly active noble metals such as platinum has been considered [4,5].

In the DUFC, the electro-oxidation of urea occurs at the anode side, and the electro-reduction reaction of oxygen occurs at the cathode side. The operating process for the DUFC in an alkaline membrane electrolyte is represented by the following reactions [3]:Cathode reaction: O_2_ + 2H_2_O + 4e^−^→ 4OH^−^     E^0^ = +0.40 V
Anode reaction: CO(NH_2_)_2_ + 6 OH^−^→ N_2_ + CO_2_ + 5H_2_O + 6e^−^     E^0^ = −0.746 V
Overall reaction: 2CO(NH_2_)_2_ + 3O_2_→ 2N_2_ + 2CO_2_ + 4H_2_O     E^0^ = +1.146 V

Multi-walled carbon nanotubes (MWCNTs) have been used as one-dimensional materials for enhancing the electron conductivity and increasing the surface area of composites for use as cathodes in DUFCs. In addition, MWCNTs are prospectively useful for increasing the porosity of catalysts. MWCNT-supported materials are widely utilized in many types of fuel cells owing to their large surface area and chemical and thermal stability in both basic and acidic media. For practical use in fuel cell catalysts, further physical and chemical treatments are needed to introduce many active sites onto the surface of MWCNTs [6,7]. Chemical activation methods include both covalent and non-covalent functionalization. Generally, non-covalent polymers can modify the delocalized π-electrons of MWCNTs when combined with a conjugated composite via van der Waals forces, π–π interactions, hydrogen bonds, and electrostatic forces. The treatment endows the active surface area of MWNCTs with functional groups such as –COOH, C-OH, and C=O– groups. Acid-activation of functionalized MWCNTs (f-MWCNTs) was implemented using concentrated nitric acid at elevated temperatures to modify the surface of carbon nanotubes in order to produce homogeneous compounds of metal or metal oxides and supported MWCNTs [8].

Compared to noble metals, transition metal oxides (TMOs), as non-platinum catalysts, have superior electrocatalytic performance for oxygen reduction reactions on cathodes of DUFCs, with many advantages such as low cost, durability, anti-poisoning ability, and stability [9]. TMO nanoparticles and MWCNTs were modified with suitable functional groups, combined with linking agents, via covalent bonding, van der Waals forces, π–π stacking interactions, hydrophobic interactions, hydrogen bonding, and electrostatic forces. Transition metals are suitable as catalysts because the cations can adopt variable oxidation states, and their oxides can be combined with other materials to achieve high tolerance and superior ORR performance [10]. However, the use of MWCNTs as catalysts for fuel cell applications is hindered by certain disadvantages, such as metal particle agglomeration, de-adhesion of the catalyst particles from the support materials, carbon degradation, and carbon corrosion [6,11]. Single-metal oxides supported by MWCNTs have been widely investigated as highly efficient bifunctional catalysts. In particular, oxides of transition metals such as Mn, Co, Ni, La, Fe, and Ce have been shown to be highly active toward the ORR [9,12,13,14]. Co_3_O_4_ nanoparticles having Co^2+^ and Co^3+^ valence states which occupy their tetrahedral and octahedral sites, respectively, are favorable for oxygen reduction reaction corresponding to changing valence states. That is conducive to the adsorption of O_2_. Therefore, Co_3_O_4_ nanoparticles, as a transition metal oxide, were combined with MWCNTs to achieve enhanced electrochemical performance as the cathode catalyst for the ORR in DUFCs and to reduce the usage of noble metals such as Pt, Pd, and Ru. The combined carbon-based cobalt oxide composites and the construction of homogeneous nano-architectures can considerably enhance the catalyst activity by producing a large catalytically active surface area and affording high conductivity, as well as affording the potential for surface modification [15,16].

Of all the tested metal materials, platinum-based catalysts exhibit the best performance in the oxygen electro-reduction reaction. However, Pt-based catalysts have not been widely used in fuel cell applications because of their high cost and scarcity. Because of this, other cost-effective non-platinum catalysts have been intensively studied to optimize the efficiency of targeted fuel cells. Mikolajczuk-Zychora et al. employed Pd nanoparticles on MWCNTs as a cathode catalyst for direct formic fuel cells [17]. The results demonstrated that the combination of Pd nanoparticles with carbon-supported materials afforded high electrocatalytic performance toward the ORR. Numerous efforts have been made to improve the catalytic performance and durability of Pd catalysts [9,10,18,19]. Carrión-Satorre et al. investigated the performance of carbon-supported palladium and palladium-ruthenium catalysts in alkaline direct ethanol fuel cells [4,20]. Pd exhibited relatively low electrochemical performance, where the electrochemical activity of Pd could be improved by employing metal oxides, which enhanced the stability and prevented degradation of the composites.

In this study, Pd nanoparticles loaded onto Co_3_O_4_ supported by multi-walled carbon nanotubes (Pd/Co_3_O_4_@MWCNT) were synthesized as a promising cathode catalyst for enhancing the electrocatalytic activity and ORR of cathodes in DUFCs. Co_3_O_4_@MWCNT and Pd/Co_3_O_4_@MWCNT were synthesized using a simple two-step hydrothermal process. The uniform dispersion of Pd nanoparticles on the Co_3_O_4_ surface was confirmed using FESEM and TEM. A Pd/MWCNT catalyst was also prepared and evaluated to investigate the effects of Co_3_O_4_ on the electrocatalytic performance of the Pd/Co_3_O_4_@MWCNT catalyst. Cyclic voltammetry (CV), rotating disk electrode (RDE), and rotating ring-disk electrode (RRDE) techniques are used to examine the ORR activity and electrochemical properties of the catalyst. This study reports the effect of Co_3_O_4_ on the composite of Pd-coated MWCNTs.

## 2. Materials and Methods

### 2.1. Materials

Cobalt(II) acetate tetrahydrate, palladium(II) chloride (PdCl_2_, 99.9%), ethylene glycol (anhydrous, 99.8%), and MWCNTs (with diameters of 50–90 nm, ≥95% carbon basis) were obtained from Sigma Aldrich (St. Louis, MO, USA). Commercial Pt/C (20 wt.%; Sigma Aldrich) was used for comparative purposes. All chemicals were used as received, without further purification.

### 2.2. Preparation of Co_3_O_4_/MWCNT Nanocomposite

First, the raw MWCNTs were pretreated with 6.0 M nitric acid and then sonicated for 2 h at room temperature, followed by magnetic stirring at 80 °C for the duration of acid treatment of the MWCNTs. Finally, the mixture was washed with DI water a few times to remove the residues, followed by centrifugation at 8000 rpm for 5 min. Subsequently, the mixture was suspended in 50 mL of ethanol under continuous magnetic stirring for 8 h at room temperature. Co(CH_3_COO)_2_ was used at different weight ratios. Note that the calculated weight ratio of the MWCNTs vs. Co(CH_3_COO)_2_ was 1:3 and 1:5, respectively. Co(CH_3_COO)_2_ (0.5 g) was added to the MWCNT suspension. Thereafter, NH_3_ (25%) aqueous solution was slowly added dropwise under continuous magnetic stirring at 45 °C. After 1 h, the mixture was placed into a Teflon-lined vessel, sealed, and maintained at 150 °C for 4 h. Finally, the sample was dried at 70 °C overnight and collected for further synthesis [21,22,23].

### 2.3. Preparation of Pd/Co_3_O_4_@MWCNT Catalyst

A modification of the experimental procedure described by Carrión-Satorre et al. was implemented herein [4]. First, chloropalladic acid was obtained by the reaction between palladium(II) chloride (0.2 M) and hydrochloric acid (0.06 M) under continuous magnetic stirring, followed by the addition of 30 mL of surfactant (ethylene glycol). After magnetic stirring for 30 min, the mixture was adjusted to pH 8 by dropwise addition of 0.5 M KOH. Subsequently, the mixture was added to 40 mg of the Co_3_O_4_@MWCNT suspension and sonicated for 2 h. Thereafter, the mixture was magnetically stirred continuously for another 2 h at 80 °C. The mixture was then placed into a Teflon-lined vessel, sealed, and maintained at 150 °C for 4 h. The mixture was centrifuged and washed with DI water and ethanol. Finally, the obtained composite was dried overnight in a vacuum oven at 70 °C. Pd@MWCNT composite was also synthesized for comparison with the same preparation of Pd/Co_3_O_4_@MWCNT catalyst, except for using the equal amount of MWCNTs instead of the Co_3_O_4_@MWCNT.

### 2.4. Characterization of the Materials

The morphologies of the prepared catalysts were examined using transmission electron microscopy (TEM, G2 F30, Tecnai, OR, USA) and scanning electron microscopy (SEM, S-4700, Hitachi). The crystalline structures of the catalysts were investigated by X-ray diffraction (XRD, Rigaku, TX, USA) using Cu-Kα radiation over the 2θ range of 20–80°. Brunauer-Emmett-Teller (BET) analysis was performed to investigate the specific surface area of the obtained samples (Autosorb iQ Station 2).

### 2.5. Electrochemical Analyses

A three-electrode system was used to investigate the electrochemical activity of the synthesized catalysts. Glassy carbon with the synthesized catalysts, a Ag/AgCl saturated electrode, and Pt wire were employed as the working electrode, reference electrode, and counter electrode, respectively. First, the glassy carbon electrode was polished with alumina powder (mean diameter of 0.3 µm), then rewashed with DI water, sonicated for 1 min, and finally dried naturally at room temperature. The catalyst ink was prepared by suspending 10.0 mg of the synthesized catalyst in a mixture of Nafion (150 µL, 5 wt. %) and isopropanol (850 µL). This suspension was sonicated for 15 min to obtain a uniform black catalyst paste. Subsequently, 6 µL of this prepared mixture of catalysts was dropped onto the surface of the glassy carbon electrode for further electrochemical tests under O_2_ atmosphere. Cyclic voltammetry was performed using a CHI 150 electrochemical workstation (Shanghai, China) with a three-electrode cell system.

### 2.6. Membrane Electrode Assembly and Fuel Cell Testing

Membrane electrode assemblies (MEAs) for single-cell direct urea fuel cells with an active working area of 5.0 cm^2^ were fabricated using FAS-30 (FuMA-Tech) membranes and carbon paper (AvCarb © MGL190 Teflon treated) fabricated by using the prepared catalyst paste. The paste was prepared by mixing 50 mg of the prepared powder in 850 µL iso-propanol/150 µL Nafion-5%. The mixture was sonicated for 15 min and magnetically stirred for 1 h. Using brushes, the paste was hand-painted onto carbon paper at a loading of ~6.0 mg cm^−2^. A Pd/C (~0.3 mg cm^−2^ loading, 40 wt. % Pd) commercial catalyst was used on the anode side, and the prepared catalysts were used on the cathode side. The anode, membrane, and cathode were subjected to hot pressing at 100 °C at 3.5 Pa for 5 min. The unit cell performance was investigated using a station VSP potentiostat-galvanostat (Biologic-Science, Seyssinet-Pariset, France) at 20 °C. The anode flow comprised 0.33 M urea solution and 1 M KOH (flow rate 10 mL min^−1^) and the cathode flow comprised oxygen only (flow rate 300 mL min^−1^).

## 3. Results and Discussion

### 3.1. Physicochemical Characterization of Co_3_O_4_@MWCNT and Pd/Co_3_O_4_@MWCNT

Figure 1 shows the X-ray diffraction patterns of the Pd/Co_3_O_4_@MWCNT, Pd/MWCNT, Co_3_O_4_ @MWCNT, and f-MWCNTs. The XRD pattern of Pd/Co_3_O_4_@MWCNT presented reflection peaks at 2θ = 41°, 46°, and 68°, corresponding to the (111), (200), and (220) planes of Pd, respectively. Most of the peaks in the XRD pattern of Co_3_O_4_@MWCNT fit well to the expected pattern, except for those of the (220) plane, and the other peaks were similar to those reported for Co_3_O_4_ at 2θ = 40°, 45°, 59°, and 65°, which correspond to the (331) (400), (511), and (440) planes, respectively [17,24]. Figure 1e,f shows the referred data which were indexed to the Co_3_O_4_ and Pd crystallite structures which corresponded to JCPDS No. 43-1003 and JCPDS No. 46-1043. As we could see, the peaks fitted well with the diffraction peaks in the obtained Pd and Co_3_O_4_ composite. This observation suggests the formation of Pd nanoparticles coated on Co_3_O_4_@MWCNT in the composite. The crystallite sizes were estimated by Debye-Scherrer equation at 2θ = 36.8° and 40.1° for Co_3_O_4_ and Pd, which were 12.3 nm and 7.8 nm, respectively. Those results relatively correspond to the sizes obtained in the TEM analysis.

The successful synthesis of the Pd/Co_3_O_4_@MWCNT composite was confirmed via TEM and SEM, as shown in Figure 2. Figure 2a presents the surface morphology of the composite comprising Co_3_O_4_ nanoparticles uniformly coated on the surface of the functionalized MWCNTs. The diameter of the MWCNTs varied from 50 to 80 nm. The surfaces of the acid-treated MWCNTs were mostly uniform without any defects, indicative of a homogeneous layer of Co_3_O_4_ particles on the MWCNTs, which afforded high uniformity without agglomeration of the particles. Figure 2b,c illustrates that the Pd/Co_3_O_4_@MWCNT composite was obtained with a relatively homogeneous distribution of the particles. However, some Pd aggregates and Co_3_O_4_ nanoparticles were still present on the surface of the MWCNTs, which could hinder the catalytic activity of the composite owing to the limited surface area. This observation is attributed to de-adhesion of the Pd nanoparticles from the Co_3_O_4_@MWCNT composite because of weak metal bonding interactions among the particles [4,25]. Figure 2d‒h presents elemental mapping of the prepared Pd/Co_3_O_4_@MWCNT as a means of reconfirming the distribution of each element in the obtained composite. The estimated particle size of Pd and Co_3_O_4_ could be obtained from TEM images. The sizes of Pd nanoparticles varied from 4.52 nm to 9.09 nm, and the Co_3_O_4_ nanoparticles varied from 9.53 nm to 16.21 nm. The mean sizes of Pd particles and Co_3_O_4_ particles were estimated as 7.62 nm and 13.18 nm from the TEM pictures, which are shown in Figure 2i,j. Furthermore, the particle size distribution of Pd/Co_3_O_4_ in Figure 2k was highly corresponding to the size distribution of the nanoparticles in the individual components. It is clear that there were distinguishable size differences of Pd particles and Co_3_O_4_ particles in the obtained Pd/Co_3_O_4_@MWCNT composite.

The high concentration of oxygen vacancies could facilitate oxygen adsorption and covalent metal oxide–nanocarbon bonding and enable improved electron transfer across the interface. The specific surface area of Pd/Co_3_O_4_@MWCNT was 190.89 m^2^ g^−1^, as obtained from the BET data presented in Figure 3. Besides, the BET specific surface area results of MWCNTs, Co_3_O_4_@MWCNT, and Pd/MWCNT were 89.8 m^2^ g^−1^, 110.2 m^2^ g^−1^, 132.1 m^2^ g^−1^, respectively. The large specific surface area of the synthesized composite facilitated the adsorption and transport of O_2_ and H_2_O during the ORR.

### 3.2. Electrochemical Characterization of Pd/Co_3_O_4_@MWCNT

Figure 4a displays the cyclic voltammograms of the prepared catalysts in 0.5 M KOH in the range of 0.2–1.4 V at a scan rate of 10 mV s^−1^. For commercial Pd/C with 20 wt.% metal loading, cathodic peaks were observed. The current density profile of Pd/Co_3_O_4_@MWCNT had two distinctive maxima of the reduction cathodic peaks at 9.93 mA.cm^−2^, which were much higher than those of Co_3_O_4_@MWCNT and Pd/C in the same investigation. The catalyst exhibited relatively good performance in the oxygen reduction reaction, and might be suitable for use as a cathode in direct urea fuels. Co_3_O_4_ nanoparticles having Co^2+^ and Co^3+^ occupying their tetrahedral and octahedral sites, respectively, are the most employed catalysts for the ORR [24,26]. The ORR corresponds to the Co^2+^ content because the Co^2+^ active surface areas are conducive to the adsorption of O_2_ and continuous electron transfer during the ORR. Co_3_O_4_, as a transition metal oxide, exhibits higher electrocatalytic activity with respect to the current density and power density of the investigated fuel cells [9]. This observation clearly indicates that the formation of a homogeneous composite enhanced the electrochemical performance due to the high active surface area of the resulting catalysts. This indicates that morphology control and surface modification are the main factors affecting the activity of the catalysts for the ORR in DUFCs [27,28].

For comparison, the polarization curves for the ORR were also recorded in 1 M KOH solution at 2000 rpm, as shown in Figure 4b. The electrocatalytic activity of the composite catalyst was higher than that of its individual components. The Pd/Co_3_O_4_@MWCNT catalyst outperformed the other catalysts owing to its higher current density and positive half-wave potential. The results illustrate that the onset potential of the composite (0.67 V) was higher than that of the commercial Pd/C catalyst (0.62 V). The current density on the composite was 9.93 mA cm^−2^ at 0.67 V, which is higher than those of the commercial Pd/C catalyst (6.53 mA cm^−2^) and Pd/MWCNT (5.03 mA cm^−2^), and is superior to that of Co_3_O_4_@MWCNT (1.62 mA cm^−2^). To gain insight into the kinetics of the ORR, the LSV data for Pd/Co_3_O_4_@MWCNT were recorded in O_2_-saturated 1.0 M KOH at various rotation speeds [29,30,31].

As shown in Figure 5, the polarization curves suggest that the measured current intensity increased with the high-speed rotation rate due to enhanced diffusion. Based on the diffusion in the kinetically limited regions, the Koutecky-Levich (K-L) plot was used to determine the electron transfer number. The K-L equation is as follows:(1)$$\frac{1}{J}=\frac{1}{J_{L}}+\frac{1}{J_{k}}=\frac{1}{Bw^{1a/2}}+\frac{1}{J_{k}}$$
B = 0.62*n*F*C_o_D_o_*^2/3^*ν*^−1/6^(2)
where *J* is the measured current density, *J_k_* and *J_L_* are the kinetic- and diffusion-limiting current density, *ω* is the electrode rotation rate, F is Faraday’s constant (96,485 C mol^−1^), *ν* is the kinetic viscosity in the electrolyte (0.01 cm^2^ s^−1^), *C_o_* is the concentration of oxygen in the electrolyte (7.8 × 10^−7^ mol cm^−1^), and *D_o_* is the diffusion coefficient O_2_ (1.8 × 10^−5^ m^2^ s^−1^). The number of electrons transferred, determined using Equations (1) and (2), was approximately 3.67 electrons on average, indicating a four-electron oxygen reduction pathway.

Figure 5c shows the RRDE curves of Pd/Co_3_O_4_@MWCNT at a rotating speed of 2000 rpm in O_2_-saturated 1.0 M KOH. The ORR is under mixed kinetic-diffusion control in the potential range between 0.5 V and 1.1 V, followed by a region where diffusion limiting currents can be observed. The ring and disk currents both increased dramatically because of the saturated oxygen environment, which is supplied to the working electrode surface area. This could result in an enhancement of the oxygen reduction reaction [32,33].

Figure 6 presents the durability test results for the commercial Pd/C catalyst and Pd/Co_3_O_4_@MWCNT composites. Fuel tolerance and stability are vital characteristics of high-performance ORR catalysts. The current density of commercial Pd/C decreased by 57%, whereas that of the Pd/Co_3_O_4_@MWCNT composite catalyst decreased to a lesser extent and retained 68% of its initial value. With regard to the long-time durability of the catalysts, the voltammograms showed high stability of the Pd/Co_3_O_4_@MWCNT composite, with no distinct current change; the initial limiting current density was 0.67 V (vs. RHE). All chronoamperometric responses demonstrate that the Pd/Co_3_O_4_@MWCNT composite possesses high durability and favorable kinetics. Therefore, the composite can be used effectively as a cathode catalyst in alkaline fuel cells.

Furthermore, the use of the prepared sample as a catholyte in the DUFC was examined [34]. Figure 7 shows the polarization and power density curves of Pd/Co_3_O_4_@MWCNT at room temperature. A current density of 13.963 mA cm^−2^ and a maximum power density of 2.792 mW cm^−2^ at 20 °C were obtained, which are relatively high compared to the values presented in other studies (Table 1). For Pd-based cathode catalysts and Pt-based catalysts, the maximum power density was achieved in some studies at approximately half of the maximum power density obtained herein. This demonstrates the promising electrocatalytic performance of the Pd/Co_3_O_4_@MWCNT composite in urea fuel cell applications. Finally, we could utilize the synthesized composite instead of commercial catalysts to reduce the cost by the dilution of Pd with non-precious metal oxides for a highly active catalyst. The durability will be investigated in long term tests to use in practical applications and large-scale utility for higher catalyst’s properties in future researches.

## 4. Conclusions

Relatively homogeneous distribution of Pd/Co_3_O_4_ on the MWCNTs was achieved in the Pd/Co_3_O_4_@MWCNT composite via a facile two-step hydrothermal process, but a few aggregates of Pd nanoparticles still remained in the synthesized composites, which could lead to a limited electrochemically active surface area for the cathode catalyst. CV evaluation of the electrochemical properties showed that the PdCo_3_O_4_@MWCNTs exhibit superior catalytic performance relative to the Pd/MWCNT and commercial Pd/C catalysts. A current density of 13.963 mA cm^−2^ and a maximum power density of 2.792 mW cm^−2^ at 20 °C were achieved. In conclusion, Pd/Co_3_O_4_@MWCNT is a prospectively effective cathode catalyst for DUFCs, affording the dilution of Pd with non-precious metal oxides.

## Figures and Tables

**Figure 1 nanomaterials-11-01017-f001:**
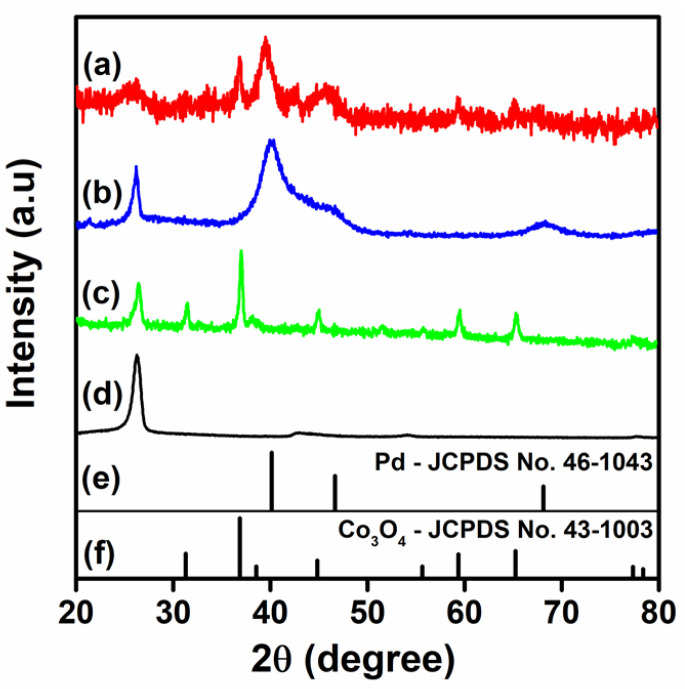
XRD patterns of: (**a**) Pd/Co_3_O_4_@MWCNT; (**b**) Pd/MWCNT; (**c**) Co_3_O_4_ @MWCNT; (**d**) f-MWCNTs; (**e**) JCPDS file of Pd; (**f**) JCPDS of Co_3_O_4_.

**Figure 2 nanomaterials-11-01017-f002:**
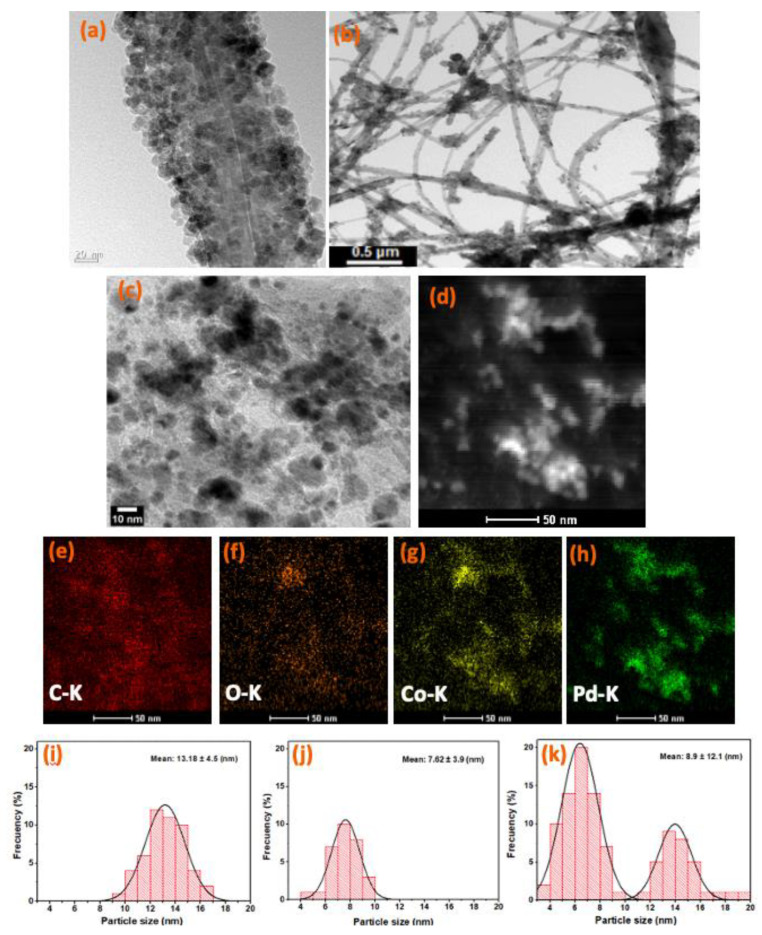
TEM images of (**a**) Co_3_O_4_@MWCNT; (**b**,**c**) PdCo_3_O_4_@MWCNT; (**d**) STEM image of Pd/Co_3_O_4_@MWCNT; (**e**‒**h**) EDS elemental mapping for Pd/Co_3_O_4_ @MWCNT composite; the size distribution of (**i**) Co_3_O_4_ NPs, (**j**) Pd NPs, (**k**) Pd/Co_3_O_4_ NPs.

**Figure 3 nanomaterials-11-01017-f003:**
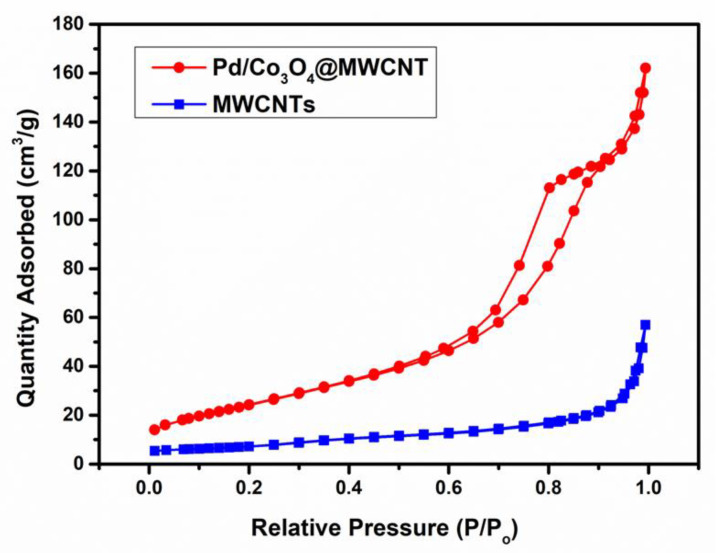
Brunauer–Emmett–Teller (BET) nitrogen adsorption–desorption isotherms of Pd/Co_3_O_4_@MWCNT and MWCNTs.

**Figure 4 nanomaterials-11-01017-f004:**
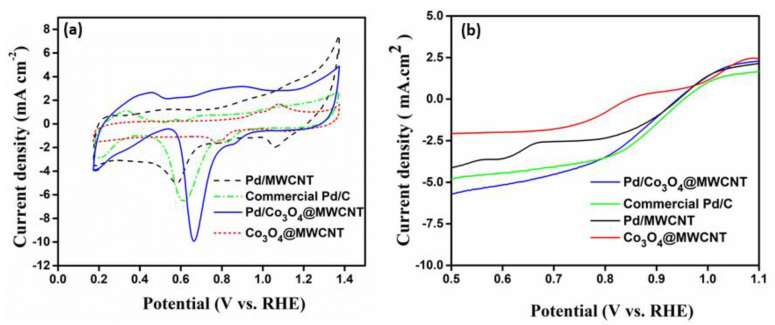
Electrochemical characteristics of the catalysts: (**a**) CV traces and (**b**) LSV curves of the Pd/Co_3_O_4_@MWCNT, Pd/MWCNT, commercial Pd/C, and Co_3_O_4_ @MWCNT catalyst at 2000 rpm in 1.0 M KOH at a scan rate of 10 mV s^−1^.

**Figure 5 nanomaterials-11-01017-f005:**
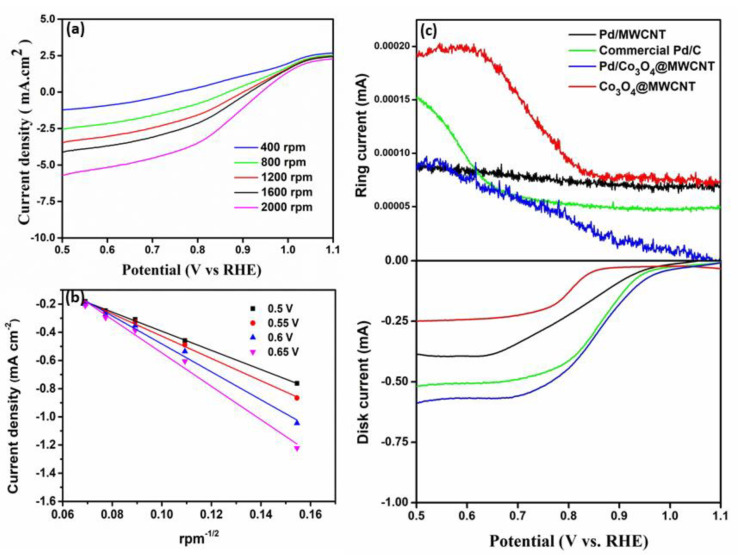
(**a**) LSV curves of Pd/Co_3_O_4_@MWCNT at different rotation rates. (**b**) K−L plots for synthesized catalysts from RDE measurements. (**c**) RRDE curves for the ORR using different electrocatalysts at 2000 rpm in O_2_−saturated 1.0 M KOH aqueous solution.

**Figure 6 nanomaterials-11-01017-f006:**
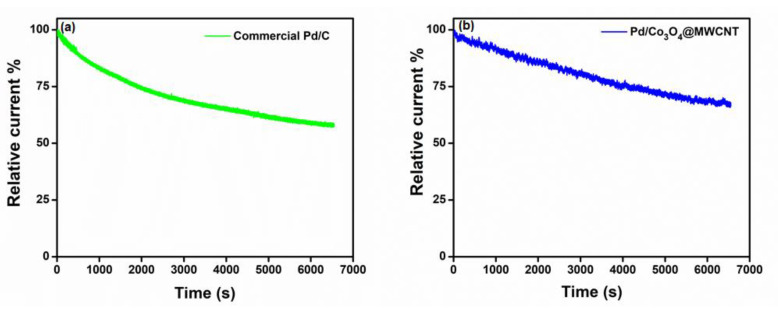
Durability test for (**a**) commercial Pd/C catalyst and (**b**) Pd/Co_3_O_4_@MWCNT with injecting 0.33 M urea in O_2_-saturated 1.0 M KOH electrolyte solution.

**Figure 7 nanomaterials-11-01017-f007:**
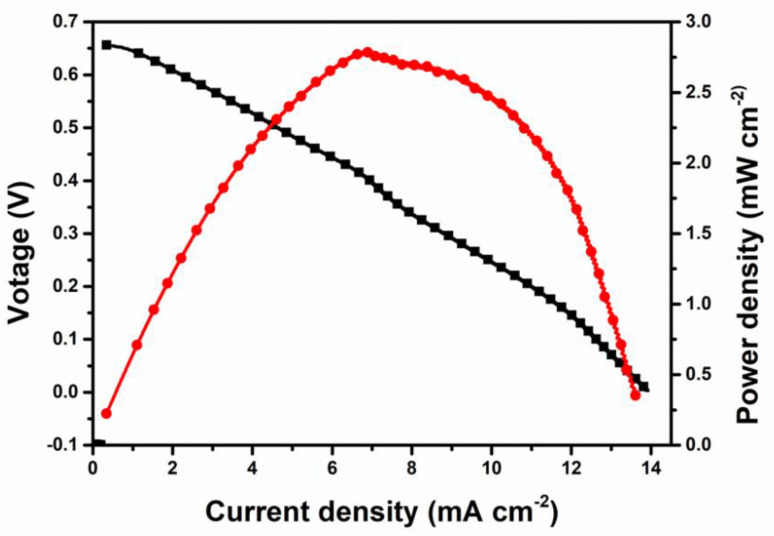
Performance of urea/O_2_ fuel cell using commercial Ni/C, FAS−30, and Pd/Co_3_O_4_@MWCNT as anode, membrane, and cathode materials, respectively.

**Table 1 nanomaterials-11-01017-t001:** Comparison of performance of different cathode catalysts in DUFC.

Cathode Catalyst	Anode Catalyst	Fuel (Anode/Cathode)	Membrane	Temperature (°C)	Maximum Power Density (mW cm^−2^)	Ref.
Pd/C	Pd-Ni/C	0.33 M Urea/O_2_	FAA (Fumasep FAA-3)	25	1.12	[19]
Pt/C	Ni/CNT	1 M Urea + 3 M KOH/O_2_	PMMA (Polymethyl-methacrylate)	25	1.6	[35]
Pt/C	NiCo/C	1 M Urea + 1 M KOH/O_2_	CEM&AEM (Cation Exchange Membranes and Anion Exchange Membranes)	20	1.4	[36]
Pt/C	Gr/Ni 3%	0.3 3M Urea + 1 M KOH/Air	AEM (Anion Exchange Membranes)	20	4.09 × 10^−3^	[7]
Mn_3_O_4_-Co_3_O_4_ @MWCNT	Ni/C 20%	0.33 M Urea + 1 M KOH/O_2_	FAA (Fumasep FAA-3)	50	0.422	[37]
Pd/Co_3_O_4_ @MWCNT	Ni/C 20%	0.33 M Urea + 1 M KOH/O_2_	FAS (Fumasep FAS-30)	20	2.792	This work

## Data Availability

Not applicable.

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
