# Peer review of "Effect of Co3O4 Nanoparticles on Improving Catalytic Behavior of Pd/Co3O4@MWCNT Composites for Cathodes in Direct Urea Fuel Cells"

_nanomaterials, 2021, doi:10.3390/nano11041017_

Round 1

Reviewer 1 Report

Title: Effect of Co3O4 Nanoparticles on Improving Catalytic Behavior of Pd/Co3O4@MWCNT Composites for Cathodes in Direct Urea Fuel Cells

Article Type: Full length article

Manuscript Number: nanomaterials-1160380

This work presents a method to synthesize Co3O4@MWCNT and Pd/Co3O4@MWCNT via a facile two-step hydrothermal process for the cathode catalyst in DUFCs. The authors claim that Pd nanoparticles loaded onto Co3O4 supported by multi-walled carbon nanotubes as a promising cathode catalyst for enhancing the electrocatalytic activity and ORR of cathodes in DUFCs.

My recommendation is that the authors carefully consider the below points, revise appropriately.

  1. My suggestion is that the authors may consider reviewing the former literature from S.J.Liu in Chem. Comm. (Chem. Commun., 2009, 4809–4811) & Energy (Energy 36 (2011)) who was the first reporter in transition metal oxides (TMOs), as non-platinum catalysts in FC cathode. It would helpful for the authors to describe the results in this article.
  2. The authors should consider more representative word in the keywords.
  3. Line 201~212; as we know the Co3O4 is an inorganic compound with two well characterized cobalt oxides which formula is sometimes written as CoIICoIII2O4 and sometimes as CoO•Co2O3. Especially, CoIIO undertakes important work for oxygen reduction in cathode. My suggestion is that the peak in XRD patterns should be marked carefully.
  4. Line 202; from the shape of line (a), (b) and (c), the particle size not only Co3O4 but also Pd are quite different in XRD determination. Could the authors show the particle size both Pd and Co3O4 in each sample?
  5. The authors should check all diffraction peaks in XRD patterns of (a), (b) and (c) which matches the JCPDS reported value and cite the number of JCPDS.
  6. Line 213~232; the authors had reported diameter of the MWCNTs varied from 50 to 80 nm. But, the particle size distribution of Pd and Co3O4 were not reported. The catalytically behaviours are Pd/Co3O4 domain that particle size distribution should be reported.
  7. Line 303~316; generally, transition metal oxide in low valence can be selected as catalyst for oxygen reduction in fuel cell cathode. Past decades the durability test for the catalyst used in FC, 200 hours durability test is basic requirement, from the time scale in Fig.7, 7000 second durability test indeed too short. If the catalyst is suitable for fuel cell that the long term test is necessary. Therefore, the authors should give us an acceptable explanation.

Author Response

Please find the Reply to the Review Report in the attached word file.

Reviewer 2 Report

The Authors present a Pd/Co3O4/MWCNTs catalyst for oxygen reduction reaction used in a direct urea fuel cell. The idea to use this kind of material is interesting, however there are many issues that shoul be taken into account before publishing this paper.

  1. The novelty of the performed research should me stated more clearly.
  2. Materials and methods should have a more detailed description - especially the part about fuel cell testing.
  3. Figure 1 should be removed from the Results section. It could be used for a graphical abstract instead.
  4.  There should be a more detailed discussion provided with the results. Right now the paper just describes the results without any discussion. The Authors should provide information why was a specific analysis used for the material and what was the outcome of it.
  5. Some analyses are performed for more materials and some are just for one type of material (e.g. XRD is performed for Pd/Co3O4/MWCNTs, Pd/MWCNT - is this a bought material or is it produced by the Authors?, Co3O4/MWCNTs, f-MWCNTs, while BET is performed only for Pd/Co3O4/MWCNTs and MWCNTs - functionalized?). It would be good if the Authors provided more results supporting their conclusions.
  6. Electrochemical results should be described with more details. Why was LSV and CV performed? What information can be extracted from the obtained results?
  7. Microscopic measurements should be changed. Figures 3a and 3b look alike in the pictures, while they are different samples, Figures 3c and 3d are named TEM while they were taken in different modes - which should be noted, the scale bar in Fig.3c is almost invisible - should be changed. Also, average particle size could be calculated from the pictures, because it is an important property in catalysis.
  8. XRD measurements should be described in more details. The Authors could estimate the average crystallite size from the patterns, which is a crucial parameter for catalysis.
  9. The acronyms in Table 1 should be explained.

Author Response

(The authors gave the same response as above.)

Reviewer 3 Report

The authors presented new important data on cathode catalyst for enhancing the electrocatalytic activity and oxygen reduction reaction at the cathode in direct urea fuel cells. Before the manuscript can be accepted for publication, minor corrections, listed below, have to be made:

Materials and Methods:

  1. “used at different weight ratios” instead of “used at different weight percentages”.
  2. “calculated weight ratio” instead of “calculated weight percentage”.
  3. “NH3 (25%) aqueous solution” instead of “NH3 (25%) solution”.
  4. “6.0 mg cm-2” instead of “6.0 mg.cm-2”.
  5. “40% wt. Pd” instead of “40% Pd”.

Table 1:

  1. “mW cm-2” instead of “mW.cm-2”.

Author Response

(The authors gave the same response as above.)

Round 2

Reviewer 1 Report

Title: Effect of Co3O4 Nanoparticles on Improving Catalytic Behavior of Pd/Co3O4@MWCNT Composites for Cathodes in Direct Urea Fuel Cells

Article Type: Full length article

Manuscript Number: nanomaterials-1160380-v2

This work presents a method to synthesize Co3O4@MWCNT and Pd/Co3O4@MWCNT via a facile two-step hydrothermal process for the cathode catalyst in DUFCs. The authors claim that Pd nanoparticles loaded onto Co3O4 supported by multi-walled carbon nanotubes as a promising cathode catalyst for enhancing the electrocatalytic activity and ORR of cathodes in DUFCs.

After reviewed the revised version, my recommendation as following:

  1. The authors should carefully check the name and symbol in full article for some of incorrect symbol. All the chemical formulas in full article should be corrected to meet IUPAC requirement. For example, line 263 ~ 267 in revised versions, both Co3O4 and Pd@Co3O4 should be Co3O4 and Pd@Co3O4.
  2. My suggestion is that the authors may consider to add more explanation in conclusion about “we could utilize the obtained composite instead of commercial one to reduce the cost. “ and “We will perform long term test to check the durability for using in practical applications and large-scale utility to achieve higher catalyst’s properties in future researches. “which were mentioned by authors in “Comment 7”.

Reviewer 2 Report

The manuscript was revised, and it looks better at the moment. But I would like to leave some of the earlier comments with answers to them along with my concerns about some parts of the manuscript, that I think that still need to be changed and revised before the publication.

1. Materials and methods should have a more detailed description - especially the part about fuel cell testing.

Answer to Comment: Thanks for the comment about Materials and methods section. For fuel cell testing procedure, we described all the detailed of the test performed in the practical system. It could be a bit different for each system, but still the same procedure for investigating the cell performance.

Comment to Answer: In the manuscript the Authors write: “Membrane electrode assemblies (MEAs) for single-cell direct urea fuel cells (…) were fabricated using FAS-30 (FuMA-Tech) carbon paper (…)” which suggests that the FAS-30 is carbon paper and not the membrane. Please correct the sentence. Another sentence: “Using brushes, the paste was hand-painted onto carbon paper at a loading of ~6.0 mg cm-2. A Pd/C (~0.3 mg cm‒2 loading, 40% wt. Pd,) commercial paper was used on the cathode side.” is also unclear. Since the Authors study materials for ORR, they should be using their material for the cathode and the commercial Pd/C paper would be used on the anode side? Please correct. During the ORR usually there are issues with water that is produced during the reaction and can block the catalysts active sites, so there are hydrophobic additives (e.g. Teflon) used for catalyst ink preparation or special hydrophobic carbon papers used. What kind of carbon paper was used for the fuel cell tests in this work?

2. Figure 1 should be removed from the Results section. It could be used for a graphical abstract instead.

Answer to Comment: Based on the reviewer comment, we have move the Figure 1 out of the Results section and put it into the Materials and Method section with its explanation.

Comment to Answer:  I do not agree with the text that “Figure 1 presents a schematic diagram of the formation of the Pd nanoparticle catalyst supported on the Co3O4@MWCNT composite using a simple two-step hydrothermal method.”. Figure 1 shows something more like I have mentioned before a graphical abstract rather than a schematic diagram of the catalyst formation. On the left side there is a nanotube, next to it is an arrow with “hydrothermal reactions” written above it and next to it there is a catalyst with attached nanoparticles and an oxygen reduction reaction on it. If the Authors want to leave the text, please adjust the figure to be in agreement with it. The schematic diagram of the catalyst formation should have such information as functionalization of the MWCNTs, what precursors were used for the formation of Co3O4 or Pd nanoparticles, etc.

3. Some analyses are performed for more materials and some are just for one type of material (e.g. XRD is performed for Pd/Co3O4/MWCNTs, Pd/MWCNT - is this a bought material or is it produced by the Authors?, Co3O4/MWCNTs, f-MWCNTs, while BET is performed only for Pd/Co3O4/MWCNTs and MWCNTs - functionalized?). It would be good if the Authors provided more results supporting their conclusions.

Answer to Comment: Thanks for the comment about analyses are performed in this study. We did perform presented analyses for all synthesized materials. However, this studies focused on investigate the electrochemical activities, so some were not presented in physical characterization sections.

In this study, the Pd@Co3O4/MWCNTs, Pd/MWCNT and Co3O4/MWCNTs are produced by us, only the raw MWCNTs were bought and oxidized by concentrated acid treatment.

We additionally carried out the BET analysis for Pd/MWCNT and Co3O4/MWCNTs as shown in Figure R2.

Figure R2. Brunauer–Emmett–Teller (BET) nitrogen adsorption–desorption isotherms of Pd/MWCNT and Co3O4/MWCNTs.

Comment to Answer: Please add in the Materials section information about the preparation of Pd/MWCNT catalyst. Additionally, the Authors now write that: “However, there were several of agglomerations of Pd@Co3O4 composite leading to hindering specific surface area of the synthesized composite.”, while this sentence is not supported by the results that they are showing. Please add the information about the BET surface for the other samples or where it can be found.  

4. Microscopic measurements should be changed. Figures 3a and 3b look alike in the pictures, while they are different samples, Figures 3c and 3d are named TEM while they were taken in different modes - which should be noted, the scale bar in Fig.3c is almost invisible - should be changed. Also, average particle size could be calculated from the pictures, because it is an important property in catalysis.

Answer to Comment: Thanks for the comment about microscopic measurements, the Figures 3a and 3b look alike in the pictures, while they are different samples. However, for SEM analysis at the same magnification, we tried to present the best images from the obtained results. Actually, the densities of the particles coated on each sample were different from each other, but the pictures are hard to distinguishing the differences. Therefore, the other analyses like EDS elemental mapping for Pd/Co3O4 @MWCNT composite could present the distribution of Pd on the prepared samples.

The estimated particle size of Pd and Co3Ocould be obtained from TEM images. The sizes of Pd particles were varied from 6.5 nm to 10.8 nm, and the Co3O particles were varied from 10.2 nm to 14.1 nm. The mean sizes of Pd particles and Co3O particles are estimated as 8.2 nm and 13.1 nm from the TEM pictures.

The scale bar in Fig.3c is modified properly as shown in the manuscript:

Comment to Answer: If you can’t see any significant differences from SEM pictures of two different samples, then I suggest removing them from the manuscript and provide TEM pictures instead, showing MWCNTs, Ce3O4@MWCNT and Pd/Ce3O4@MWCNT. Especially, when the Authors state that: “The estimated particle size of Pd and Co3Ocould be obtained from TEM images. The sizes of Pd particles were varied from 6.5 nm to 10.8 nm, and the Co3O particles were varied from 10.2 nm to 14.1 nm. The mean sizes of Pd particles and Co3O particles are estimated as 8.2 nm and 13.1 nm from the TEM pictures.”, but from Figure 3c how the Authors could distinguish Pd nanoparticles from Co3O4 nanoparticles? And it would be better to show a histogram of the particle size distribution. The Authors should write what TEM modes were the pictures taken in.

5. XRD measurements should be described in more details. The Authors could estimate the average crystallite size from the patterns, which is a crucial parameter for catalysis.

Answer to Comment: The crystallite sizes would be estimated from Debey-Scherrer equation at 2θ = 36.8â—¦ and 40.1â—¦ for Co3O4 and Pd are 12.3 and 7.8 nm as calculation from the obtained data, respectively. These particles size is smaller than the sizes obtained in TEM analysis.

Comment to Answer: Thank you for the answer, but could you also put this information in the manuscript? It supports the microscopic measurements. If the results differ from the microscopic measurements it could suggest either formation of aggregates or formation of amorphous oxide on the surface of the particles that could be observed in the microscope.
